# Protein Tyrosine Phosphatase PRL-3: A Key Player in Cancer Signaling

**DOI:** 10.3390/biom14030342

**Published:** 2024-03-12

**Authors:** Haidong Liu, Xiao Li, Yin Shi, Zu Ye, Xiangdong Cheng

**Affiliations:** 1Zhejiang Cancer Hospital, Hangzhou 310022, China; liuhd@zjcc.org.cn; 2Hangzhou Institute of Medicine (HIM), Chinese Academy of Sciences, Hangzhou 310018, China; 3The Second Clinical Medical College, Zhejiang Chinese Medical University, Hangzhou 310053, China; xiaoli@zcmu.edu.cn; 4Department of Biochemistry, Zhejiang University School of Medicine, Hangzhou 310058, China; yinshi@zju.edu.cn; 5Key Laboratory of Prevention, Diagnosis and Therapy of Upper Gastrointestinal Cancer of Zhejiang Province, Hangzhou 310022, China; 6Zhejiang Provincial Research Center for Upper Gastrointestinal Tract Cancer, Zhejiang Cancer Hospital, Hangzhou 310022, China

**Keywords:** protein phosphatases, phosphatase of regenerating liver-3 (PRL-3), cancer, drug resistance

## Abstract

Protein phosphatases are primarily responsible for dephosphorylation modification within signal transduction pathways. Phosphatase of regenerating liver-3 (PRL-3) is a dual-specific phosphatase implicated in cancer pathogenesis. Understanding PRL-3’s intricate functions and developing targeted therapies is crucial for advancing cancer treatment. This review highlights its regulatory mechanisms, expression patterns, and multifaceted roles in cancer progression. PRL-3’s involvement in proliferation, migration, invasion, metastasis, angiogenesis, and drug resistance is discussed. Regulatory mechanisms encompass transcriptional control, alternative splicing, and post-translational modifications. PRL-3 exhibits selective expressions in specific cancer types, making it a potential target for therapy. Despite advances in small molecule inhibitors, further research is needed for clinical application. PRL-3-zumab, a humanized antibody, shows promise in preclinical studies and clinical trials. Our review summarizes the current understanding of the cancer-related cellular function of PRL-3, its prognostic value, and the research progress of therapeutic inhibitors.

## 1. Brief Introduction of Protein Phosphatases

In the 1950s, protein phosphorylation was discovered as an important post-translational modification (PTM) that determines protein function [1]. It has a crucial function in multiple cellular activities, such as cell growth, proliferation, differentiation, migration, motility, programmed cell death, and metabolism [2]. According to estimates, phosphorylation regulates 30% of human genome-encoded proteins [3]. Protein kinases are responsible for transferring phosphate groups from ATP to substrate proteins, while protein phosphatases play a role in removing phosphate groups from phosphoproteins and transferring them to water molecules. Phosphatases and kinases are crucial enzymes involved in cellular signal transduction, regulating protein phosphorylation [4].

Based on historical classification, protein phosphatases primarily comprise two major families: protein serine/threonine phosphatases (PSTPs) and protein tyrosine phosphatases (PTPs). The PSTP family consists of approximately 45 members, whereas the PTPs superfamily encompasses over 100 members [5]. There are currently 241 human active and 13 inactive phosphatases discovered in total, which have been reclassified into 16 superfamilies based on their structure and function (https://depod.bioss.uni-freiburg.de/index.php (accessed on 21 January 2024)). 

PTPs are highly modular, with about 75% of PTPs containing an additional domain or motif on top of the core catalytic domain, providing a greater diversity of functions and regulation targets [6]. The PTPs family possesses the ability to dephosphorylate both tyrosine and serine residues. The catalytic domain of all these PTPs includes a phosphate-binding loop (P-loop) or protein-tyrosine phosphatase (PTPase), which consists of an active site motif HCXXGXXR (where X can be any amino acid). In spite of significant differences in the HCXXGXXR section, the shared amino acids of cysteine and arginine lead to a consistently maintained structure of the P-loop [7]. In addition to the P-loop, the acid loop (known as WPD-loop) is also conserved in PTPs, undergoing conformational changes during the catalytic process. (Figure 1A) [8]. The PTPs can be broadly classified into several distinct subfamilies such as Receptor-type tyrosine-protein phosphatase (R-PTP), tyrosine-specific non-receptor PTPs (NR-PTPs), VH1-like dual-specificity protein phosphatases (DUSPs), Phosphatase of Regenerating Liver (PRL) based on the amino acid sequences of their catalytic domains [6,9,10,11].

## 2. PRL Family

The PRL family, a unique class of DUSPs, comprises three members: PRL-1, PRL-2, and PRL-3 (also known as PTP4A1, PTP4A2 and PTP4A3, respectively) [8]. Ranging around 20 kDa, with PRL-2 at 167 amino acids and PRL-1 and PRL-3 at 173 amino acids, these phosphatases share significant sequence homology. The amino acid sequences of the three PRLs display significant similarity. PRL-1 and PRL-2 exhibit an 87% homology, PRL-1 and PRL-3 share a 79% homology, and PRL-2 and PRL-3 have a 76% homology. (Figure 1B) [12,13].

*PRL* genes in humans are found on separate chromosomes, with *PRL-1*, *PRL-2*, and *PRL-3* situated on chromosomes 6q12, 1p35, and 8q24.3 correspondingly. Small phosphatases PRLs are identified as important enhancers of intracellular magnesium levels [14,15,16,17]. They are key enzymes involved in cellular signal transduction, which demonstrates potential carcinogenic activity [18,19].

*PRL-1* was to be recognized as an immediate-early gene that is significantly increased in the regenerating liver of rats [20]. The expression of *PRL-1* mRNA is observed in multiple human tissues, especially in the small intestine, lung, oviduct, testis, gallbladder, T-cells and adipocytes [21,22]. The protein functions as a signaling molecule within cells, exerting regulatory effects on numerous cellular processes, such as cell proliferation and migration [23,24]. The protein may also be involved in cancer development and metastasis [25,26].

Similarly, elevated amounts of *PRL-2* mRNA can be found in almost all human tissues, with the exception of taste buds and extensively specialized fibrocartilage tissues [21]. The extensive distribution of *PRL-1* and *PRL-2* mRNA suggests their potential involvement in fundamental biological processes shared by the majority of tissues and cell types [27]. PRL-2 is required for vascular morphogenesis, hematopoietic stem cell self-renewal and angiogenic signaling [28,29,30]. Overexpression of PRL-2 in mammalian cells conferred a transformed phenotype, which suggested its role in tumorigenesis. According to recent research, PRL-2 facilitates the degradation of PTEN, thus promoting tumor occurrence and development [31,32].

In contrast to the ubiquitous expression of PRL-1 and PRL-2 in various tissues, PRL-3 exhibits selective expression in specific organs and cancer cells, making it an appealing target for cancer treatment [33,34,35]. Increased levels of *PRL-3* have been documented in cases of liver cancer, breast cancer, ovarian cancer, papillary renal cell carcinoma, and various other conditions (Figure 2). This elevation could potentially impact the disease’s prognosis [36,37].

## 3. Regulatory Mechanisms of PRL-3

Although being the last discovered among the PRLs, PRL-3 has garnered significant attention and has been extensively studied. Multiple reports have indicated that the regulation of PRL-3 expression occurs at various levels, encompassing DNA, RNA, and protein levels (Figure 3) [38,39].

In most cases, the human *PRL-3* gene has a solitary copy on chromosome 8, near the long arm, with 9613 nucleotides in length. Increased levels of *PRL-3* have been documented in cases of liver cancer, breast cancer, ovarian cancer, papillary renal cell carcinoma, and various other conditions [40,41,42,43]. This elevation could potentially impact the disease’s prognosis. Initially, it was believed that this amplification of copy number was accountable for the elevated expression of PRL-3 in cancerous conditions. Nevertheless, multiple studies indicate that there is an absence of a substantial correlation between amplification of the *PRL-3* gene and expression of mRNA, suggesting that the expression of PRL-3 might be tightly controlled during transcription as well [44,45].

The initial evidence supporting this claim is based on the discovery that the famous oncogene p53 functions as a transcriptional regulator of *PRL-3*. The interaction between p53 and the *PRL-3* genomic region has been noted, leading to the activation of its transcription in human and mouse cell lines [46]. Subsequent identification of other transcription factors, such as Snail, myocyte enhancer factor 2C (MEF2C), signal transducers and activators of transcription 3 (STAT3), signal transducer and activator of transcription 5A (STAT5A) and Nucleolar RNA helicase 2 (DDX21), along with their respective functional promoter binding sites in the *PRL-3* gene, further underscored the complexity of *PRL-3* transcriptional regulation [45,47,48,49,50,51]. All of them demonstrate remarkable specificity and capability in inducing *PRL-3* expression. The pre-mRNA of *PRL-3* consists of 5 exons, and when exon 4 undergoes alternative splicing, it produces two distinct transcripts, leading to the formation of two isoforms of PRL-3 protein [52]. Unlike the full-length PRL-3 protein, the spliced variant, which comprises only 148 amino acids, does not possess phosphatase activity [8]. Poly(C)-binding protein 1 (PCBP1) recognizes and binds to three GC-motifs (GCCCAG) present in the 5’-UTR of *PRL-3* mRNA. This RNA-binding protein PCNP1 serves multiple functions, including mRNA stabilization and translation suppression [53]. The interaction between PCBP1 and the GC motif leads to the inhibition of PRL-3 protein synthesis, while PCBP1-AS1, which is oriented in the opposite direction to PCBP1, enhances the generation of PRL-3 protein [53,54].

The regulation of the PRL-3 is also influenced by PTMs. PTMs play a crucial role in maintaining physiological homeostasis by regulating protein structure, destination, activity, stability, and function, thus contributing to protein destruction or turnover [55]. Currently, the known PTMs for the PRL-3 protein include Ubiquitin, prenylation, oxidation, and palmitoylation [51,56,57]. It is generally believed that PRL-3 undergoes PTM at the C-terminus to bind to the cell membrane and exert biological activity, and then interacts with downstream effectors [58].

## 4. PRL-3 in Cancer

PRL-3 was initially discovered in 2001 as the only gene that exhibited a significant increase in expression in metastases originating from colorectal carcinomas (CRCs) while remaining undetectable in normal colon epithelia affected by liver cancer [56]. Subsequently, more studies confirmed this phenomenon and found that PRL-3 was highly expressed in other primary and metastatic tumors, including gastric cancer [59,60,61], colorectal cancer [62,63], breast cancer [42,64], liver cancer [43,65], intrahepatic cholangiocarcinoma [66], lung cancer [67,68], esophageal cancer [69], nasopharyngeal carcinoma [70], uveal melanoma [71], ovarian and cervical carcinoma [72,73]. PRL-3 promotes cancer cell proliferation, migration, metastasis, and angiogenesis through multiple signaling pathways (Figure 4).

### 4.1. Proliferation and Tumorigenesis

The capacity for sustained proliferation and metabolic alterations are significant attributes of cancer cells [74,75]. The growth rate of HEK293 cells expressing PRL-3 was increased compared to the inactive PRL-3 mutant (C104S) [36]. This accelerated growth rate was mitigated by the inhibition of PRL-3 using PTPase inhibitors, thereby confirming the necessity of phosphatase activity for the PRL-3-mediated augmentation of cell proliferation. Moreover, the injection of B16 melanoma cells overexpressing PRL-3 into in vivo xenograft mouse models resulted in a significant three-fold rise in tumor volume when compared to the control cells [76].

Recent studies have shown that the presence of PRL-3 enhances the invasion and proliferation of malignant cells. Moreover, elevated levels of *PRL-3* have a strong correlation with poor Overall Survival (OS) and Progression-free Survival (PFS) in patients with breast cancer and glioblastoma [42,77]. In contrast, the inhibition of natural PRL-3 through RNA interference significantly hindered the growth of different cancer cell lines, including ovarian, lung, gastric, colorectal, and leukemia cancers [34,41,60,63].

Multiple studies have provided evidence suggesting that PRL-3 facilitates cell proliferation by activating Src kinase, promoting STAT3 signaling, and modulating cell cycle regulators such as cyclin D1, CDK2, STAT5, and AKT [78,79,80,81]. Previous research has indicated that the overexpression of PRL-3 in HEK293 cells triggers the activation of Src kinase by inhibiting C-terminal Src kinase (Csk), which is a suppressor of Src [82]. There is evidence that SW480 colon cancer with heightened Src activity and low Csk expression possesses lower PRL-3 expression compared with SW620 cells. The activation of Src triggers a number of downstream signaling pathways that stimulate cell proliferation [78]. STAT3 signaling cascades are also found to be involved in PRL-3-induced cell proliferation [79,80]. STAT3 triggers the upregulation of various microRNAs, including miR-29c, miR-21, miR-17, and miR-19a, promoting cellular proliferation [83,84]. In addition, PRL-3 promotes cell cycle progression and enhances the anti-apoptotic mechanism of tumor cells to achieve drug resistance by upregulating cyclin D1 and CDK2, as well as activating STAT5 and AKT [81].

The nuclear factor-κB (NF-κB) pathway is another signaling pathway downstream involved in the promotion of cell proliferation by PRL-3. NF-κB, an extensive protein complex, is present in most animal cell varieties and regulates DNA transcription [85]. Enhancement of cellular proliferative ability in LoVo colon cancer cells is achieved through upregulation of intermediate conductance calcium-activated potassium channel protein 4 (KCNN4) expression in an NF-κB-dependent manner due to PRL-3 overexpression [86]. In addition, USP4, which is elevated in gastric and rectal cancer, interacts with PRL-3 and increases the stability of PRL-3, promoting NF-κB signaling and cell viability [51,60]. Celecoxib, a promising candidate for anticancer therapy, upregulates PTEN protein expression in mouse hepatoma tissues while downregulating NF-κB and PRL-3 protein expression, ultimately attenuating liver cell proliferation [87].

Aurora kinase A (AURKA) plays a role in the formation and stability of microtubules at the spindle pole during chromosome separation. PRL-3 enhances the ubiquitination and degradation of AURKA in a phosphatase-dependent manner. It was found that PRL-3-induced G2 m arrest was associated with reduced expression of AURKA [88]. The proliferation of cells is controlled by the regulation of the cell cycle. According to research, knockdown of the *PRL-3* gene by shRNA resulted in decreased expression of downstream Stathmin, inhibited cell proliferation, and induced G2/M arrest and cell apoptosis [89], while another study demonstrated that silence of PRL-3 reduced cell migration and cell proliferation but had no detectable effect on the cell cycle [38].

### 4.2. Migration, Invasion and Metastasis

In general, most cells carry out their specialized functions within their organs within a limited range of motion. However, cancer cells have developed the ability to spread from their original organ to other areas of the body through a phenomenon known as metastasis. Cell–cell adhesion and cell adhesion to extracellular matrix (ECM) are required for the acquisition of invasive and motile behaviors [90]. Over the last couple of years, numerous research has indicated a connection between heightened PRL-3 expression and heightened severity of cancer as well as its transfer ability. Multiple groups have verified that PRL-3 exhibited a significant increase in liver metastases of CRC, as well as in secondary CRC lesions discovered in the lung, brain, ovary, peritoneum, and lymph nodes [34,91,92]. Furthermore, the analysis of clinical statistics indicated that high PRL-3 expression appears to be associated with increased liver and lung metastasis in colorectal cancer, suggesting that PRL-3 expression may play a role in CRC metastasis [62,92]. In HEK293 cells, PRL-3 has the ability to decrease the tyrosine phosphorylation of integrin β1 while promoting the activation of ERK1/2. The activation of ERK1/2 stimulated by PRL-3 can be eliminated, and the motility and invasion of LoVo cells can be eradicated by depleting integrin β1 in vitro [92]. The expression of PRL-3 mRNA was also increased in almost all metastatic lesions derived from CRCs [39,93]. The heightened expression of PRL-3 is also associated with elevated invasion of the lymphatic and venous systems, metastasis to lymph nodes and peritoneum, and an escalation in tumor stage [34]. Similarly, the downregulation of PRL-3 inhibits migration and invasion of lung cancer cells through RhoA and mDia1 [94]. On the contrary, some reports have also surfaced indicating the absence of a substantial contribution of PRL-3 to the metastasis and proliferation of cancer cells [95].

The exact molecular basis for the increased spread of metastasis caused by PRL-3 remains largely unknown. Several signaling pathways, including PI3K/Akt, P38, JAK/STAT, ERK, integrin/Src, and Rho family GTPases, have been documented as being involved in facilitating certain migration and metastasis effects of PRL-3 (Figure 2).

The PI3K/Akt pathway is a significant oncogenic pathway that is commonly overactive in human cancers, playing a role in tumor formation, such as proliferation, invasion, and migration [96,97,98]. The PRL-3 protein activates the PI3K/Akt signaling pathways when overexpressed [99]. Activation of PRL-3 initiates a positive feedback loop involving AKT, p38, TGFβ1, and FAK, resulting in increased phosphorylation of FAK and facilitating the proliferation, migration, and adhesion of HCC cells [43]. Downregulation of the PI3K/Akt signaling pathway can significantly inhibit tumor activity, and simultaneously inhibiting the WNT/β-catenin pathway can enhance its effects in vitro and in vivo [100]. Expression of PRL-3 also enhances PIK3C3-BECN1-dependent autophagy in an ATG5-dependent manner, therefore promoting tumor growth [101].

Some reports suggest that activated p38 MAPK contributes to cancer cell drug resistance, migration, and invasion, whereas other sources claim that it acts as a tumor suppressor that regulates cell death [102]. Under stressed conditions, PRL-3 functions as a direct phosphatase for p38 MAPK. PRL-3 enhances tumor cell adaptation to the hypoxic stress tumor microenvironment and facilitates tumor lung metastasis by negatively regulating p38 MAPK activity [67,72].

STATs possess both SH2 and SH3 domains, enabling their interaction with peptide segments that bear phosphorylated tyrosine residues. Upon phosphorylation, STATs undergo polymerization, adopting an activated transcriptional activator conformation. Subsequently, STATs translocate into the nucleus, where they can engage with target genes and facilitate their transcription [103]. These pivotal proteins assume a vital function in signal transduction and the activation of transcription. IL6 promotes STAT3-dependent transcriptional upregulation of PRL-3, which in turn re-phosphorylates STAT3 and aberrantly activates STAT3 target genes [48,104]. PRL-3 overexpressed in classical Hodgkin lymphoma can inhibit the production of IL-13 cytokines and enhance STAT6 signaling, increasing cell migration and vitality [105].

It has been shown that matrix metalloproteases (MMPs) are highly expressed in many cancer cases and are associated with cancer progression, invasion, metastasis, and immune suppression [106]. PRL-3 is found to promote the proliferation, invasion, and migration of glioma cells by increasing the activity of ERK/JNK/MMP7 in vitro and in vivo [77]. Additionally, the expression of PRL-3 is significantly correlated with the expression of various MMPs, such as MMP2 and MMP9 [107,108].

### 4.3. Inducing Angiogenesis

PRL-3’s association with vascular invasion in hepatocellular carcinoma points to its role in angiogenesis [43,109]. Studies demonstrate that PRL-3, acting downstream of the VEGF/MEF2C pathway, recruits and enhances angiogenesis in endothelial cells in vitro and promotes tumor angiogenesis in vivo [47,110]. Furthermore, PRL-3 upregulates pERK and Rho expression and promotes their activity, facilitating VEGF expression and accelerating angiogenesis and distant metastasis [37,110]. Meanwhile, the induction of EGFR by PRL-3 was associated with enhanced cell proliferation, migratory properties, and tumorigenicity [111]. The activation of EGFR by PRL-3 leads to the transcriptional downregulation of protein tyrosine phosphatase 1B (PTP1B), resulting in the inhibition of EGFR activation [111]. Inactivating PRL-3 downregulated VEGF signaling by reducing the phosphorylation of ERK1/2 [112].

### 4.4. Promoting Focal Adhesion

PRL-3 is involved in various cancer-related functions, including but not limited to metastasis, proliferation, and angiogenesis. A variety of phenotypic characteristics are influenced by cell-matrix interactions, including gene regulation, cytoskeletal structure, differentiation, and cell growth [113]. PRL-3 plays a crucial role in cell adhesion and proliferation by influencing the expression levels of integrin, matrix, and Ezrin [114]. Deletion of the *PRL-3* gene from murine colorectal tumors results in impaired colony formation, spheroid formation, migration, and adhesion [115]. In mammalian cells, PRL-3 interacts with integrin α1 and integrin β1, transmembrane receptors facilitating cell–cell and cell-extracellular matrix interactions. This interaction activates the integrin/Src pathway, a key player in epithelial-mesenchymal transition, focal adhesions, and aiding in cellular migration [116,117]. PRL-3 effector pathway in metastasis involves the integrin/Src pathway, which plays a crucial role in epithelial-mesenchymal transition and focal adhesions [118,119,120]. Additionally, apart from the involvement of Akt and integrin/Src pathways, it has been reported that Rho family GTPases also participate in the regulation of PRL-3-induced metastasis. Overexpression of PRL-3 in embryonic stem cells derived from endometrioma increases the expression of RhoA, RhoC, ROCK1, and MMP9, promoting cell migration and invasion [107].

### 4.5. Other Functions

PRL-3 upregulation in colon cancer cells and primary fibroblasts induces telomere structural abnormalities, telomere deprotection, DNA damage response, chromosomal instability, and senescence, contributing to tumor progression [121]. Additionally, heightened PRL-3 phosphatase activity in the healthy intestinal epithelium disrupts intestinal cell equilibrium, increasing vulnerability to PRL-3-mediated inflammation or mutation, which may lead to tumor development [122]. Moreover, PRL-3 has been shown to interact with cyclin and CBS domain divalent metal cation transport mediator (CNNM) to regulate the intracellular concentration of calcium and magnesium plasma [123,124]. Interestingly, PRL-3’s role in promoting H+ extrusion and acid addiction via stimulating lysosomal exocytosis enhances cancer cell survival in an acidic tumor microenvironment [125].

The dynamic and complex nature of PRL-3’s role in tumor proliferation, invasion, and metastasis involves multiple signaling pathways, necessitating the activation of different effector proteins.

## 5. The Relationship between PRL-3 and Drug Resistance

Chemotherapy, radiotherapy, and targeted therapy resistance remain the main challenges for cancer treatment [126]. Cancer resistance can be roughly divided into intrinsic (primary) resistance or acquired (secondary) resistance [127]. Throughout the course of long-term chemotherapy, cancer cells undergo evolution and may acquire multidrug resistance (MDR), leading to a substantial reduction in the effectiveness of cancer treatment and negatively impacting patients’ survival and quality of life [126]. The majority of chemotherapy, radiotherapy, and immunotherapy have the potential to elevate intracellular reactive oxygen species (ROS) levels, which are responsible for cancer cell damage [128,129]. Compared with non-MDR cancer and normal cells, the levels of ROS and the activity of clearance/antioxidant enzymes are usually increased in drug-resistant cancer cells [130]. Several studies have indicated that PRL-3 decreases intracellular ROS levels and induces overexpression of glycolysis enzymes and molecules, contributing to enhanced tumor cell proliferation and invasion [131,132,133]. Silencing PRL-3 in colon cancer cells has been shown to lead to a ROS-dependent DNA damage response and senescence, indicating a potential link between PRL-3 and acquired resistance [121]. Further investigation is necessary to ascertain if PRL-3 augments cell intrinsic resistance by diminishing ROS levels, thereby enabling them to acquire acquired resistance through adequate exposure time.

Paclitaxel (PTX), a microtubule-stabilizing anticancer chemotherapeutic, is one of the most common therapeutic commonly used drugs. The mechanism underlying resistance to PTX is primarily caused by alterations in α-tubulin and β-tubulin [134]. PRL-3 interacts with α, β, and γ-tubulin, suggesting its involvement in PTX resistance [135]. Integrin-mediated focal adhesions play a crucial role in cell adhesion, migration, and therapy resistance in cancer [136,137]. The expression of PRL-3 increases the focal adhesion of cells to the extracellular matrix, making them more resistant to drug treatment [33].

Extracellular vehicles (EVs), including exosomes, microvesicles, oncosomes, and microparticles, are associated with anticancer drug resistance [138]. EVs induce cancer cell resistance by transferring specific cargos that affect drug efflux and regulate signaling pathways related to epithelial-mesenchymal transition, autophagy, and metabolism [139]. PRL-3 antigen detection on cell surfaces and EV outer membranes may be linked to cell drug resistance and serve as a potential target for cancer therapy [140].

In addition, PRL-3 also promotes the formation of grossly hyperdiploid and multinucleated cancer cells known as polyploid giant cells (PGCCs), which typically appear in tumor tissue after chemotherapy, forming a stem cell-like pool that promotes cell survival, chemotherapy resistance, and tumor recurrence [141,142].

## 6. Drug Discovery

Chemotherapy, aimed at impeding cancer cell growth and division, often lacks specificity for cancer cells, causing harm to normal tissues [143]. Targeting PRL-3, which is overexpressed in certain cancer tissues, provides an opportunity to enhance cancer treatment specificity [116]. Due to the role of PRL-3 in cancer progress, multiple drugs have been screened that can inhibit PRL-3 at the molecular level, effectively impeding the activity of PRL-3 in vivo and in vitro (Table 1).

Pentamidine, the initial pharmaceutical agent identified as effective against PRL-3, demonstrates the capacity to impede the function of PTPs and restrain the proliferation of human cancer cells [144]. Subsequent research has revealed that certain naturally occurring compounds derived from plants, including rhodanine, bioflavonoids, anthraquinones, and curcumin, possess the ability to inhibit the activity of PRL-3 [145,146,147,148,149,150]. Furthermore, thienopyridone and its derivatives (such as JMS-053 and NRT-870-59), which act as selective PRL inhibitors, have been shown to suppress the growth and migration of cells overexpressing PRL-3 [80,151,152,153,154]. Despite the fact that recent advancements have found that several new chemicals can inhibit the activity of PRL-3, there remains room for improvement in terms of enhancing the specificity, stability, and solubility of these compounds. Furthermore, given the potential for adverse side effects and toxicity associated with chemical compounds, it is imperative to conduct further investigations on these drugs before considering their application as PRL-3-targeted inhibitors in clinical cancer therapy [155].

In 2019, PRL-3-zumab, a First-in-Class humanized antibody drug, demonstrated the ability to bind to the surface of PRL-3 in a manner consistent with the classical antibody-dependent cell-mediated cytotoxicity (ADCC) action or antibody-dependent cell phagocytic tumor elimination pathway [140,156]. Subsequently, PRL-3-zumab has been proven to reduce tumor relapse in the ‘tumor removal’ animal model [142] and has been confirmed to have anti-tumor activity in the PDX model [157]. As a therapeutic mAb, PRL-3-zumab has demonstrated a strong safety profile in clinical trials, presenting a promising avenue for targeted therapy against PRL-3 [158].

The most recent study has demonstrated that nanobodies designed to target PRL-3 have the ability to disrupt PRL-3 phosphatase activity and inhibit the interaction between PRL-3 and CNNM3 by binding to the active site of PRL-3 [159]. These anti-PRL-3 nanobodies have the potential to serve as a valuable tool for further exploration of PRL-3.

**Table 1 biomolecules-14-00342-t001:** Representative drug for PRL-3.

Year	Drug	Discovery	Reference
2002	Pentamidine	The first reported drug has anticancer activity via inhibiting PTPs	[144]
2006	Biflavonoids	The first natural products reported to have inhibitory effects on PRL.	[145]
2008–2017	Thienopyridone, JMS-053, NRT-870-59	Selective PRLs inhibitor.	[80,151,154]
2016	PRL-3-zumab	A humanized antibody drug against PRL-3	[156]
2023	anti-PRL-3 nanobodies	The first alpaca-derived single-domain antibodies against PRL-3	[159]

## 7. Discussion

The dysregulation of kinases and phosphatases, crucial in maintaining cellular homeostasis, contributes to the initiation and progression of various diseases, including cancer [160,161]. Several protein phosphatases have been found to be dysfunctional prognostic markers in a variety of cancer contexts, often indicating a higher-grade disease or an advanced disease [98,162,163,164]. PRL-3, an important member of the phosphatase family, has emerged as a key player in multiple signaling pathways in the past two decades, affecting disease pathogenesis, tumor occurrence, and progression, and is also related to prognosis [165,166]. It is clear that PRL-3 signaling plays an important role in the pathogenesis and development of a wide array of human diseases. Overexpression of PRL-3 in cancers leads to diverse effects, including sustained proliferative signaling, replicative immortality, genome instability, mutation, resistance to cell death, and angiogenesis.

PRL-3’s multifaceted roles in tumor cell proliferation, movement, invasion, and metastasis involve complex mechanisms. While significant progress has been made in understanding PRL-3’s functions, further research, including knockout studies of specific regulators and effectors, is necessary to unravel its precise role in specific signaling pathways.

Crucially, PRL-3’s expression in various cancer types and absence in normal tissues make it an attractive therapeutic target. PRL-3 inhibitors have shown promise in inhibiting tumors, and their combination with anti-tumor drugs enhances therapeutic efficacy [116]. PRL-3 inhibitors will become a viable treatment option for cancer patients with further research and clinical trials. Importantly, the development of monoclonal antibodies, such as PRL-3-zumab, provides a novel and safe therapeutic approach, presenting a potential breakthrough in clinical cancer therapy.

In conclusion, our evolving understanding of PRL-3’s diverse functions and the potential of PRL-3 inhibitors in human diseases opens the door to more specific targeted therapeutic strategies. It will be possible to develop more specific targeted therapeutic strategies in the future if we have a better understanding of the multifunctions and complex regulatory mechanisms of PRL-3.

## Figures and Tables

**Figure 1 biomolecules-14-00342-f001:**
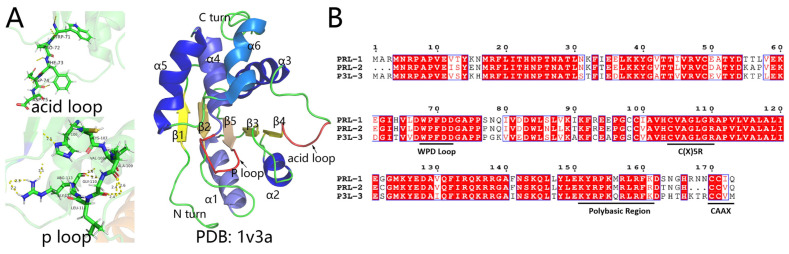
The structure and amino acid sequence of PRLs. (**A**) Structure of the acid loop, P loop, and PRL-3 (PDB:1v3a). (**B**) Aligning PRLs including PRL-1, PRL-2, PRL-3.

**Figure 2 biomolecules-14-00342-f002:**
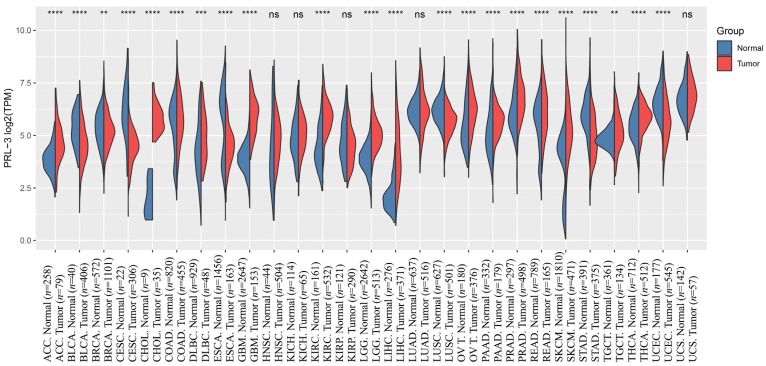
The expression distribution of PRL-3 gene in tumor and normal samples in TCGA and GTEx. The abscissa represents different tumor tissues, and the ordinate represents the expression distribution of genes, with different colors representing different groups. ns means *p* > 0.05, ** *p*  <  0.01, *** *p*  <  0.001, **** *p*  < 0.0001, asterisks (*) stand for significance levels. The raw count data were normalized using the Trimmed Mean of M-values (TMM) method and comparative analysis through the Student’s *t*-test. (ACC: adenoid cystic carcinoma, BLCA: bladder urothelial carcinoma, BRCA: Breast Invasive Carcinoma, CESC: Cervical Squamous Cell Carcinoma and Endocervical Adenocarcinoma, CHOL: Cholangiocarcinoma, COAD: Colon Adenocarcinoma, DLBC: Lymphoid Neoplasm Diffuse Large B-cell Lymphoma, ESCA: Esophageal Carcinoma, GBM: Glioblastoma Multiforme, HNSC: Head and Neck Squamous Cell Carcinoma, KICH: Kidney Chromophobe, KIRC: Kidney Renal Clear Cell Carcinoma, KIRP: Kidney Renal Papillary Cell Carcinoma, LGG: Brain Lower Grade Glioma, LIHC: Liver Hepatocellular Carcinoma, LUAD: Lung Adenocarcinoma, LUSC: Lung Squamous Cell Carcinoma, OV: Ovarian Serous Cystadenocarcinoma, PAAD: Pancreatic Adenocarcinoma, PRAD: Prostate Adenocarcinoma, READ: Rectum Adenocarcinoma, SKCM: Skin Cutaneous Melanoma, STAD: Stomach Adenocarcinoma, TGCT: Testicular Germ Cell Tumors, THCA: Thyroid Carcinoma, UCEC: Uterine Corpus Endometrial Carcinoma, UCS: Uterine Carcinosarcoma).

**Figure 3 biomolecules-14-00342-f003:**
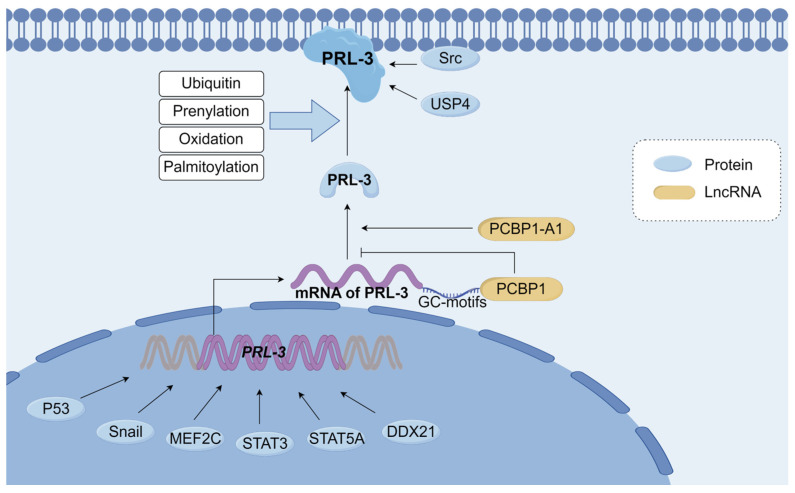
Overview of Regulatory mechanisms of PRL-3. Transcriptional regulators such as P53, Snail, MEF2C, STAT3, STAT5A, and DDX21 can promote *PRL-3* gene expression. PCBP1 recognizes the GC-motifs of *PRL-3* mRNA, suppressing the translation of *PRL-3* mRNA. PCBP1-AS1 promotes the production of PRL-3. PRL-3 undergoes ubiquitin, prenylation, oxidation, and binding to the cell membrane, exerts biological activity, and then interacts with downstream effectors. Src, USP4 interacts with PRL-3 and increases the stability of PRL-3. MEF2C: myocyte enhancer factor 2C, STST3: signal transducers and activators of transcription 3, STAT5A: signal transducer and activator of transcription 5A, DDX21: Nucleolar RNA helicase 2, PCBP1: poly(C)-binding protein, PCBP1-A1: polycytosine binding protein 1 antisense1, USP4: Ubiquitin-specific protease 4. This Figure was created using images from FigDraw.

**Figure 4 biomolecules-14-00342-f004:**
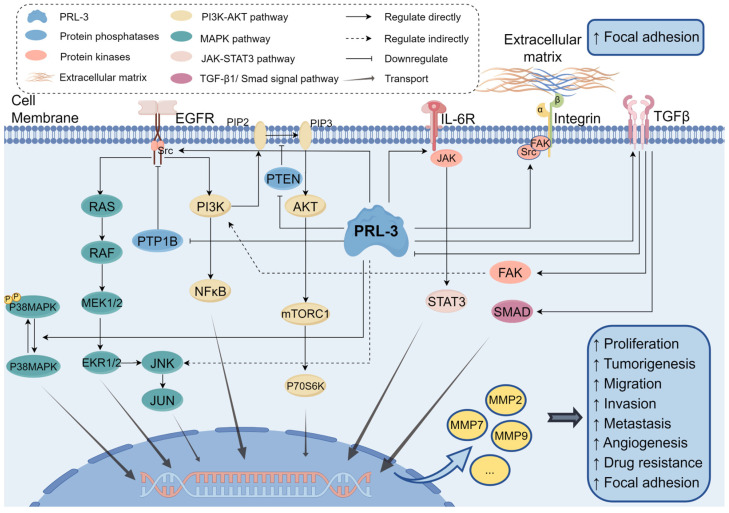
An overview of signaling pathways regulated by PRL-3 discussed in this review. PRL-3 promotes tumorigenesis by activating the PI3K-AKT pathway, MAPK pathway, JAK-STAT3 pathway, and TGF-β1/Smad signal pathway. PRL-3: Phosphatase of Regenerating Liver-3, PTP1B: protein tyrosine phosphatase 1B, PTEN: Phosphatidylinositol 3,4,5-trisphosphate 3-phosphatase and dual-specificity protein phosphatase PTEN, EGFR: epidermal growth factor receptor, IL-6R: Interleukin-6 receptor, TGFβ: transforming growth factor-β, JNK: Jun N-terminal kinase, SMAD: small mother against decapentaplegic, MMP: matrix metalloproteases. This Figure was created using images from FigDraw.

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
