# Peer review of "Protein Tyrosine Phosphatase PRL-3: A Key Player in Cancer Signaling"

_biomolecules, 2024, doi:10.3390/biom14030342_

Round 1

Reviewer 1 Report

Comments and Suggestions for Authors

This review provides a well-written synthesis of current knowledge and research progress on Phosphatase of regenerating liver 3 (PRL-3), particularly its implications in cancer development, progression, and treatment. It perfectly aligns with the journal's focus on impactful biomolecules. To further enhance reader accessibility and knowledge exchange, I recommend the following additions:

1. Visualize PTP Domain Structures: Include a figure illustrating the domain structures of protein tyrosine phosphatases (PTPs), specifically highlighting Phosphatases of Regenerating Liver (PRLs). This would provide readers with a visual context for understanding PRL-3 within the broader PTP family.

2. Illustrate PRL-3 Regulation: Another figure depicting the key mechanisms regulating PRL-3 expression and activity would be highly beneficial. This would visually consolidate the text-based discussion and enhance reader comprehension.

3. Enhance Figure 1 Transparency: For Figure 1, consider including details on sample numbers, data normalization procedures, and the specific statistical methods employed. This will increase transparency and allow readers to better evaluate the data presented.

4. Cross-reference Figure 2: Ensure the current Figure 2 is explicitly referred to and integrated within the paragraph discussing "PRL-3 in cancer" subtitle. This will improve the flow of information and guide readers to relevant visual insights.

These additions will strengthen the review's visual communication and accessibility, facilitating knowledge exchange and understanding for a wider audience.

Comments on the Quality of English Language

The following issues have been identified in the current version:

  1. Phrase fragmentation: Some phrases require modification to become complete sentences.
  2. Punctuation: Several punctuation marks need correction or adjustment for clarity.
  3. Grammar: Some grammatical errors require revision for improved accuracy.

Additionally:

  • Gene names should be italicized, while protein names should not.
  • Consider further proofreading before resubmitting to ensure accuracy and clarity.
  • Having a native English speaker review the document could further enhance the writing and eliminate any remaining errors.

The attached file contains annotated error examples for your reference.

Reviewer 2 Report

Comments and Suggestions for Authors

The manuscript entitled “Protein Tyrosine Phosphatase PRL-3: A Key Player in Cancer Signaling“ by Haidong Liu et al. summarizes the results connecting PRL-3 with cancer. There are several points concerning the manuscript that should be considered. 

1. The article summarizes data from many studies that were mainly conducted with cell lines. It is difficult to get an overall picture. I do not expect a review article to simply list all published data. Rather, an attempt should be made to create a new conceptual frame, perhaps limited to some aspects of the biology of PRL-3. Only this strategy makes a review article interesting and valuable. The present manuscript is bulky and difficult to read. The reader gets lost in all the different results.

2. Many aspects are discussed in the article without the data already giving a clear picture. Here some examples: The regulation of PRL-3 is possibly regulated by posttranslational modifications, but  “ the specific impact of PTM on PRL-3 function is still unidentified and requires further research” (pge 4). It seems that PRL-3 has an impact on cell division, but “more research is needed” (page 6). “The exact molecular basis for the increased spread of metastasis caused by PRL-3 remains largely unkown” (page 6). “Detailed mechanism and direct substrates (for PRL-3) remains areas for further investigation” (page 8). “Further investigation is necessary to ascertain if PRL-3 augments cell intrinsic restistance by diminishing ROS levels”. (page 8). There is no need for a review article if important aspects are still unknown. 

3. As far as inhibitors are concerned, the gold standard is specificity. I doubt that the compounds listed in Table 2 are PRL-3-specific. If they are not, the entire chapter of the article should be deleted.

4. Table 1 is difficult to understand because of the many abbreviations. It is not clear what the point of this table is in the article. Showing the different modular structures of protein phosphatases would probably be a better idea. For Fig. 1, it is not clear what the message is. For Fig. 2, it would be interesting to know what are the differences between this figure and the signaling figures published by Duciel (J. Mol. Biol. 431, 2019). 

Reviewer 3 Report

Comments and Suggestions for Authors

In this review, the authors analysed the role of  PRL-3 in cancer and its involvement in proliferation, migration, and metastasis. They also discussed the advances of PLR-3

Inhibitors and humanized antibody for clinical application.

Comments:

Overall, the research subject of this review is of interest. However, a profound English editing revision is required (just an example, line 98-106 the entire sentence is not clear and need to be re-phrase).

     1)       Line 36, Reference is missing

     2)       Some Acronyms need to be checked

     3)       Line 228: “numerous researchers…” but just one Ref.

     4)       Fig 2 is too general and required more details. In addition, the colour of single molecules is confusing.

     5)       4.4 “Focol adhesion” need to be corrected.

Comments on the Quality of English Language

A profound English editing revision is required 

Round 2

Reviewer 1 Report

Comments and Suggestions for Authors

The revision has demonstrably improved the manuscript's quality and accessibility for readers, particularly through the inclusion of very good schematic illustrations. Further improvements could be made by referring  Table 1 within the text to ensure a clear and cohesive understanding for the reader. Additionally, a final proofread before submission is recommended to catch any remaining errors.

Author Response

We would like to thank the reviewer for providing thoughtful, constructive and positive comments and suggestions on our manuscript (biomolecules-2858825). All modifications made in the text were shown in bule.

Table.1 adds a lot of confusion to literature reading, so we deleted Table.1 and adjusted the corresponding paragraphs in the exist line 58-62.

Reviewer 2 Report

Comments and Suggestions for Authors

The revised manuscript entitled “Protein Tyrosine Phosphatase PRL-3: A Key Player in Cancer Signaling“ by Haidong Liu et al. has been improved, but most of my comments and suggestions were not considered. 

1. Fig. 1 is very helpful by showing the structure and domains of PRL enzymes. However, the authors keep Table 1 that is not very understandable. 

2. The manuscript still has a 2 pages chapter about PRL inhibitory compounds that are probably all for most of all not PRL-specific. In addition, the evaluation method is not clearly described, i.e. has it been shown that these compounds inhibit PRL-3 enzyme activity ?

Reviewer 3 Report

Comments and Suggestions for Authors

The new version of the manuscript is improved.

Author Response

We would like to thank the reviewer for providing thoughtful, constructive and positive comments on our manuscript. We made some modifications to the article and deleted some parts in the exist line 59-62, 387-395, 402-411